Investigation of spatiotemporal evolution and driving mechanisms of carbon sources and sinks in dry-hot valleys under extreme climate events: a case study of the Nu River dry-hot valley

Sun Haojun
Zhang Shaoyun zhangshy56@mail2.sysu.edu.cn
Liu Shanshan
Zhang Liping
1 School of Soil and Water Conservation, Southwest Forestry University , Kunming , Yunnan , China
2 Key Laboratory of Ecological Environment Evolution and Pollution Control in Mountainous & Rural Areas of Yunnan Province, Southwest Forestry University , Kunming , Yunnan , China
3 Zhanyi Karst Ecosystem Observation and Research Station, National Forestry and Grassland Administration , Qujing , Yunnan , China
Phairuang Worradorn
Electronic publication date: 2025 Nov 3
Publication date: 2025
Volume: 13
Electronic Location ID: e20268
Received 2025 May 23; Accepted 2025 Sep 30
Copyright: ©2025 Sun et al.
Copyright year: 2025
Copyright holder: Sun et al.
License: This is an open access article distributed under the terms of the Creative Commons Attribution License, which permits unrestricted use, distribution, reproduction and adaptation in any medium and for any purpose provided that it is properly attributed. For attribution, the original author(s), title, publication source (PeerJ) and either DOI or URL of the article must be cited.
License URL: https://creativecommons.org/licenses/by/4.0/

Keywords: Nu River Valley, Extreme climate events, Threshold response mechanism, Remote sensing, Google Earth Engine, NPP

Funding: Yunnan Provincial Basic Research Special Fund 202401CF070079 Yunnan Provincial First-Class Discipline Open Fund in Soil and Water Conservation and Desertification Control SBK20240008 National Natural Science Foundation of China 42401005 Yunnan Provincial College Students’ Innovation and Entrepreneurship Training Program S202410677042 This work was supported by the Yunnan Provincial Basic Research Special Fund (No. 202401CF070079), the Yunnan Provincial First-Class Discipline Open Fund in Soil and Water Conservation and Desertification Control (No. SBK20240008), the National Natural Science Foundation of China (No. 42401005) and the Yunnan Provincial College Students’ Innovation and Entrepreneurship Training Program (No. S202410677042). The funders had no role in study design, data collection and analysis, decision to publish, or preparation of the manuscript.

==============================
Under global climate change, the rising frequency of extreme weather events profoundly affects ecosystem carbon cycles. However, in the ecologically fragile dry-hot valleys of Southwest China, the response of carbon source-sink dynamics to these extremes remains unclear, which hinders effective regional carbon management. This study investigates the Nu River dry-hot valley, using the Google Earth Engine platform to process multi-source remote sensing and meteorological data from 2001–2024. We established a framework of 15 extreme climate indices and applied Sen-Mann-Kendall trend analysis, Pearson correlation, and threshold regression models to explore the spatiotemporal evolution of carbon dynamics and their non-linear response to extreme climate. Our results show that: (1) The region’s carbon sink capacity displayed a fluctuating but overall increasing trend with significant spatial heterogeneity; areas of substantial increase were concentrated in the southern, low-altitude zones. (2) Extreme climate events triggered non-linear carbon cycle responses by altering hydrothermal conditions. The synergy of high temperatures and intense, short-duration precipitation weakened the carbon sink, whereas dispersed rainfall alleviated drought stress and enhanced carbon fixation. (3) Both extreme temperature and precipitation indices showed clear regulatory thresholds, above which their effects were significantly amplified; this reveals that the carbon cycle in the dry-hot valley is highly sensitive to extreme events and exhibits distinct threshold-driven responses. This research provides a theoretical basis for the mechanisms regulating carbon flux in dry-hot ecosystems under a variable climate and offers crucial scientific support for optimising regional pathways to carbon neutrality and implementing climate-adaptive management.

Introduction

Against the backdrop of global climate change, the frequent occurrence of extreme climate events has had a profound impact on terrestrial ecosystem carbon cycling processes (Zhou, Yu & Zhang, 2023). The Sixth Assessment Report of the Intergovernmental Panel on Climate Change (IPCC) explicitly states that extreme heat, droughts, and heavy precipitation events have become the new normal over the past five decades, posing a severe challenge to the carbon balance (Frank et al., 2015; Piao et al., 2019). As a core component of the terrestrial ecosystem carbon cycle, the sensitivity of carbon source–sink dynamics to extreme climate has become a frontier topic in global change ecology (Yuan et al., 2024; Li et al., 2023; Wang et al., 2025).

In recent years, extensive research has been conducted on the variations in carbon flux and their climatic drivers. At the global scale, studies have clearly shown that extreme climate events profoundly alter the carbon sink function of terrestrial ecosystems by disrupting processes such as photosynthesis, respiration, and water use efficiency (Pan et al., 2020), often exhibiting pronounced non-linear response characteristics (Frank et al., 2015; Reichstein et al., 2013). Drought is widely considered one of the most destructive climatic factors at present, with its suppressive effect on net primary productivity (NPP) and net ecosystem productivity (NEP) far exceeding its impact on respiration (Piao et al., 2019). In fact, the ecosystem’s response to such climatic shocks depends on the stability of its structural carbon pools (i.e., biomass, litter, and soil), which is shaped by local topography and edaphic conditions (Arasa-Gisbert et al., 2018; Bisht et al., 2022) and ultimately forms the basis of the region’s ecological resilience (Anderson, 1971; Fartyal, Bargali & Bargali, 2025). Particularly in water-limited arid ecosystems, soil moisture acts as the core driver mediating carbon-water fluxes (Kannenberg et al., 2024). The impact of extreme precipitation, however, differs significantly between arid and humid regions; it can enhance carbon fixation in dry areas but may exacerbate carbon loss in wet areas due to soil disturbance (Yin et al., 2023). As research has progressed, the focus has shifted from single events to the synergistic and lagged effects of compound extremes (Li & Wu, 2024). For instance, the combined impact of drought and heatwaves can disproportionately amplify the negative effects on vegetation productivity (Zhou et al., 2024), and the delayed influence of extreme temperatures on vegetation carbon flux is also receiving increasing attention (Huang et al., 2023). These advancements have significantly improved our understanding of the complexity of carbon-climate coupling. Despite the growing clarity of these macro-scale patterns, knowledge gaps persist at the regional scale, especially in topographically complex and ecologically fragile areas. The dry-hot valley region of Southwest China, a marginal zone highly sensitive to climate change, exhibits an unstable carbon budget that further magnifies the ecosystem’s vulnerable response to climatic disturbances (Fan et al., 2020). Although existing studies have revealed that the carbon sink potential in this region is strongly regulated by altitude, land use, and ecological restoration projects (Lv et al., 2023; Cheng & Lü, 2024; Li et al., 2024), and that the threat from extreme climate is intensifying, this research has largely focused on linear analyses of mean annual climate conditions. There remains a lack of systematic, quantitative identification of the non-linear regulatory pathways triggered by extreme climate events, particularly the critical climatic thresholds that can induce shifts in response patterns (Zhao et al., 2024b). Furthermore, the complex topography within the dry-hot valleys leads to high spatial heterogeneity in hydrothermal conditions, yet how this translates into a specific pattern of carbon sink sensitivity under the influence of extreme climates, and the underlying mechanisms thereof, remain to be elucidated (Wang et al., 2021). A more in-depth quantitative assessment of the synergistic effects of different types of extreme climate events and their combined impact on the carbon cycle is also required (Huang & Zhai, 2024; Li et al., 2021). These unresolved scientific questions limit our comprehensive understanding of the carbon dynamics in this unique ecosystem and constrain the scientific rigour and precision of climate-adaptive management strategies (Feng et al., 2024; Pu et al., 2022).

Given this context, and considering the current lack of exploratory research on the dynamic patterns and threshold characteristics of carbon sink responses to extreme climate in montane, ecologically sensitive areas like dry-hot valleys, this study focuses on a typical dry-hot valley area within the Nu River basin. By integrating 15 extreme climate indices with remote sensing products for carbon flux, we constructed an annual gridded dataset to investigate the following three key scientific questions: (1) Between 2001 and 2024, did the evolution of carbon sources and sinks in the study area exhibit significant spatiotemporal trends, and did these patterns show local clustering or heterogeneity? (2) Do extreme temperature and precipitation events exhibit combined effects or synergistic amplification mechanisms, and are their disturbances to carbon sink capacity characterised by regional variability and time lags? (3) Are there critical thresholds or turning points in climatic variables that drive abrupt shifts in carbon flux responses, and how do these non-linear mechanisms manifest spatially and temporally? To address these questions, we hypothesize that the carbon sink dynamics in the Nu River dry-hot valley exhibit distinct spatiotemporal patterns and are governed by non-linear threshold responses to extreme climatic forcing. A systematic investigation of these questions will help to deepen the understanding of the mechanisms governing carbon cycle responses to extreme climate disturbances in dry-hot regions, providing a model basis and identifying sensitive factors to support the formulation of regional-scale carbon neutrality strategies.

Study Area and Data Sources

Study area and ecological background

The Nu River dry-hot valley is located in the middle and lower reaches of the Nu River Basin in Southwest China, covering an area of approximately 2,107.13 km2 (Fig. 1). It is a archetypal example of the region’s dry-hot valleys. Situated on the western flank of the Hengduan Mountains, the area is characterised by a deeply incised landscape resulting from its pronounced alpine gorge topography. The Nu River flows from north to south through the entire territory; the valley floor is narrow, flanked by steep mountains, and features significant altitudinal variation. This topography results in distinct vertical zonation of both climate and ecosystems. Owing to its enclosed terrain and the combined influence of the Southwest Indian Monsoon and the Tibetan Plateau air mass, the Nu River dry-hot valley has developed a unique hot and dry climate. This environment is defined by high temperatures, low precipitation, and intense evaporation, and is prone to frequent extreme climate events (Chen, Xu & Li, 2019; Peng et al., 2018; Yin et al., 2023). The region’s mean annual temperature is high, typically ranging between 20–25 °C, with extreme maximum temperatures exceeding 40 °C and prolonged periods of hot weather in the summer. Annual precipitation is low, approximately 600–800 mm, and is concentrated between June and October, creating distinct wet and dry seasons. Due to strong solar radiation and atmospheric circulation, evaporation in the Nu River dry-hot valley far exceeds precipitation, with annual potential evaporation reaching over 3,000 mm. This creates a hydro-thermal imbalance that severely constrains vegetation growth and recovery (Zheng et al., 2018). The local soils are predominantly dry red soils, alluvial deposits, and sandy substrates. These soils are characterised by shallow profiles and poor water-retention capacity, which exacerbates the ecosystem’s vulnerability and makes land degradation and soil erosion particularly prominent issues (Zhang et al., 2022). The boundary data for the study area were derived from the 2020 dataset of dry valley distribution in Southwest China, published by the Institute of Mountain Hazards and Environment, Chinese Academy of Sciences. For this paper, the Nu River dry-hot valley section was selected for investigation (Fan et al., 2020).

Figure 1 Overview of the study area in the Nu River Arid Valley Region.

Data sources and processing

Carbon source and sink estimation

Carbon sources and sinks are vital indicators for assessing an ecosystem’s carbon budget. Net Primary Productivity (NPP), a commonly used metric, is effective for evaluating carbon sink capacity (Zou et al., 2022). The NPP data used in this study were from the MODIS (MOD17A3HGF) dataset, which provides global-scale annual NPP data. It has a temporal resolution of 1 year, a spatial resolution of 500 m, and its units are kg C/m2/yr, making it suitable for analysing trends in vegetation productivity. To ensure the reliability of this data for trend analysis in our study area, we conducted a preliminary cross-validation against the independent GLASS NPP product (sourced from the National Earth System Science Data Center, National Science & Technology Infrastructure of China; http://www.geodata.cn, 2001–2020). The results demonstrated a strong correlation (Pearson’s r > 0.6) in the interannual NPP trends for the Nu River basin between the two datasets, confirming the suitability of the MODIS data for the analytical objectives of this study. The 2001–2024 MOD17A3HGF dataset used in this study was acquired and exported via the Google Earth Engine (GEE) platform for subsequent analysis.

Calculation of extreme climate indices

A total of 10 meteorological stations are located within and surrounding the Nu River dry-hot valley. Daily maximum temperature, minimum temperature, and precipitation data from 2001 to 2024 were obtained from the China Meteorological Data Service Centre (http://data.cma.cn). Using the Python 3.11 platform, stations with more than 30 days of missing data in any given year were excluded. Gaps of one to two days were filled using the average of the adjacent days, while longer continuous gaps were filled using the long-term average for the same period from other years. Extreme climate indices such as the diurnal temperature range (DTR), summer days (SU25), growing season length (GSL), frost days (FD0), and ice days (ID0) were calculated. These indices have been widely applied in previous research and are highly interpretable (see Table 1 for specific definitions) (Xu et al., 2020). The calculated extreme precipitation and temperature indices were interpolated using the inverse distance weighting (IDW) method in ArcGIS 10.8. This method assigns weights based on the principle of spatial proximity and does not require complex parameterisation. Consequently, it provides a stable and reasonable spatial distribution pattern under conditions of limited sampling, effectively reducing the uncertainty of the interpolation results. It demonstrates robustness and applicability in study areas with sparse station coverage.

Table 1 Overview of extreme climate indices.

Index type	Index name	Abbreviation	Definition	Unit	
Extreme precipitation indices	Annual total precipitation	PRCPTOT	Cumulative precipitation on days with precipitation ≥1 mm	mm	
Moderate rain days	R10	Number of days with daily precipitation >10 mm	d (days)	
Heavy rain days	R25	Number of days with daily precipitation >25 mm	d (days)	
Very wet days	R95p	Total precipitation on days when daily precipitation >95th percentile	mm	
Extremely wet days	R99p	Total precipitation on days when daily precipitation >99th percentile	mm	
Max 1-day precipitation	RX1day	Maximum 1-day precipitation	mm	
Max 5-day precipitation	RX5day	Maximum consecutive 5-day precipitation	mm	
Simple daily intensity index	SDII	Annual total precipitation/Number of days with precipitation ≥1 mm	mm/d	
Extreme temperature indices	Summer days	SU25	Number of days when daily maximum temperature (TX) >25 °C	d (days)	
Annual min value of Tmin	TNn	Minimum value of annual daily minimum temperature (TN)	°C	
Annual max value of Tmax	TXx	Maximum value of annual/monthly daily maximum temperature (TX)	°C	
Warm spell duration index	WSDI	Days per year with at least 6 consecutive days of Tmax >90th percentile	d (days)	
Cold spell duration index	CSDI	Days per year with at least 6 consecutive days of Tmin <10th percentile	d (days)	
Diurnal temperature range	DTR	Difference between daily maximum (TX) and minimum temperature (TN)	°C	
Icing days	ID0	Number of days when daily maximum temperature (TX) ≤0 °C	d (days)	

Research Platform and Methodology

Trend analysis

A linear regression method was employed to characterise the temporal trend of carbon sources and sinks (NPP) in the Nu River dry-hot valley. The annual rate of change in NPP is represented by the slope of the regression line fitted using the ordinary least squares method (Han & Han, 2025; Xu et al., 2020). A positive slope indicates an increasing trend in NPP, suggesting an enhancement of the regional carbon sink capacity. Conversely, a negative slope indicates a decreasing trend, suggesting an increase in regional carbon emissions. The linear regression equation is as follows: (1) y=bx+a

where y is the time series dataset for NPP, b is the slope of the NPP change, x is the time sequence from 2001 to 2024, and a is the intercept.

Correlation analysis

Correlation analysis is an important method for revealing the degree of association between different variables. In this study, a pixel-based Pearson correlation coefficient was used to analyse the correlation between NPP and extreme climate indices in the Nu River dry-hot valley (Cui et al., 2024). The formula for the Pearson correlation coefficient is: (2) Rxy=∑i=1nxi−x ¯yi−y ¯∑i=1nxi−x ¯2 ∑i=1nyi−y ¯2

where x and y denote the NPP and extreme climate index respectively, xi and yi represent the values for year i, and barx, y ¯ are the multi-year means. The coefficient Rxy ranges from −1 to 1: values closer to 1 indicate a strong positive correlation, while values closer to −1 indicate a strong negative correlation.

Sen’s slope and Mann-Kendall (Sen-MK) trend test

The Sen-Mann-Kendall (Sen-MK) trend test is a non-parametric statistical method commonly used to analyse trends in time series data. This method determines the existence of a trend by comparing the rank order of data points within the time series (Frank et al., 2015; Luo et al., 2023). Compared to traditional parametric statistical methods, the Sen-MK test does not rely on assumptions about the data distribution, making it suitable for various types of time series data. The formulas for this method are as follows: (3) β=mediumXj−Xij−i,∀j>i

where Xj and Xi are observations at time j and i, respectively. A positive β indicates an upward trend, and a negative β indicates a downward trend. The Mann-Kendall test statistic S is defined as: (4) S= ∑j=2n ∑i=1j−1sgnxj−xi

where the sign function sgn(xj −xi) equals 1 if xj > xi, 0 if xj = xi, and −1 if xj < xi. The significance of the trend is assessed using the standardized test statistic Z: (5) Z=S−1nn−12n+518

where S is the statistic calculated above and n is the number of observations. The significance of the trend can be determined by comparing ∣Z∣ with the critical value from the standard normal distribution at a given significance level. In this study, the 24-year (2001–2024) NPP distribution dataset was analysed using a Sen-MK trend test procedure in a Python 3.11 environment. The resulting images were then processed and statistically analysed using the ArcGIS 10.8 platform.

Threshold regression model

To accurately identify potential threshold effects in the influence of climatic factors on NPP, this study employs a segmented linear regression model (Hansen, 2000; Toms & Lesperance, 2003). This model is capable of capturing non-linear relationships where a structural change between the independent and dependent variables occurs at a critical point. For a single unknown threshold (γ), the model can be expressed as a unified linear equation through the introduction of an indicator function, I(⋅): (6) Yi=β0+β1Xi+β2Xi−γ∗IXi>γ+ϵi

In this model, Yi represents NPP, Xi is the climatic factor, and the terms β1 and β1 + β2 represent the respective slopes of the relationship before and after the threshold. As the threshold γ is unknown, this study utilises a grid search method. The procedure involves trimming 15% of the samples from each end of the data distribution and then iterating through all possible remaining values as candidate thresholds. The optimal threshold is identified as the value that minimises the model’s residual sum of squares (RSS). The identification of this point, therefore, signifies the critical value at which the pattern of the climatic factor’s influence on NPP undergoes a shift.

Results

Temporal variation of NPP in the Nu River dry-hot valley

During the study period (2001–2024), the total annual net primary productivity (TNPP) in the Nu River dry-hot valley exhibited notable fluctuations but followed a generally increasing trend. In certain years, TNPP was significantly impacted by extreme climate events, which led to considerable declines (Fig. 2).

Figure 2 Total NPP trend from 2001 to 2024.

The TNPP of the study area began at 7.40 × 104 kg C yr−1 in 2001 and followed an upward trend until 2008, when it peaked at 8.03 × 104 kg C yr−1. This increase was attributed to favourable temperature and precipitation conditions that promoted vegetation photosynthesis. A significant decrease occurred in 2010, with TNPP falling to 7.28 × 104 kg C yr−1, a decline closely associated with a combination of frequent high-temperature events and unfavourable precipitation patterns that suppressed the vegetation’s net carbon fixation capacity.

Following this nadir, TNPP recovered to 7.93 × 104 kg C yr−1 in 2011, indicating a degree of ecosystem resilience. However, this recovery was short-lived, as TNPP decreased again to 7.39 × 104 kg C yr−1 in 2012. This drop was potentially related to a lower annual minimum temperature (TNn) and a higher number of ice days (ID0), with these low-temperature extremes likely shortening the effective growing season. Between 2013 and 2016, TNPP remained relatively stable, fluctuating around a mean of 7.70 × 104 kg C yr−1. Another marked decrease to 7.26 × 104 kg C yr−1 occurred in 2018, possibly due to comprehensive climatic stresses. Subsequently, TNPP gradually recovered from 2019 to 2021, reaching 8.12 × 104 kg C yr−1 by 2022. Although extreme precipitation events were numerous that year, their uniform distribution may have provided sufficient moisture for growth. After a slight decrease in 2023 (8.09 × 104 kg C yr−1), TNPP reached its highest recorded value of 8.40 × 104 kg C yr−1 in 2024.

Despite these significant interannual fluctuations driven by extreme climate events, vegetation NPP in the Nu River dry-hot valley still showed an overall increasing trend over the past two decades. This suggests adaptation and resilience of the regional vegetation, potentially enhanced by other drivers such as CO2 fertilization effects or localised ecological improvements. Nevertheless, the potential long-term impacts of extreme temperatures, droughts, and altered precipitation patterns on the region’s carbon fixation capacity warrant continued attention (Fig. 3).

Figure 3 Spatial distribution of NPP from 2001 to 2024.

Vegetation NPP response to extreme climate events

Correlation between NPP and extreme climate indices

To investigate the response of net primary productivity (NPP) to extreme climate events in the Nu River dry-hot valley, this study employed a pixel-based Pearson correlation analysis. We examined the relationship between the interannual variation of NPP and various extreme climate indices for each grid cell (Fig. 4). The analysis revealed that extreme precipitation events generally had a significant positive correlation with NPP. Among these, very heavy precipitation days (R95p) showed the strongest positive correlation, with an average coefficient (R) of 0.703. The maximum 1-day precipitation (RX1day) also displayed a strong positive correlation. These results indicate that in this water-stressed region, moderate extreme precipitation can effectively replenish soil moisture and alleviate seasonal drought, thereby promoting photosynthesis and NPP accumulation. For example, in years with less than 800 mm of total annual precipitation, a 10-day increase in heavy precipitation days (R10) corresponded to an approximate 4.2% increase in regional average NPP. In contrast, some extreme temperature indices demonstrated a clear inhibitory effect on NPP. The warm spell duration index (WSDI) was significantly negatively correlated with NPP density (R = −0.184), suggesting that prolonged heat stress suppresses net carbon fixation by increasing evapotranspiration, soil water loss, and vegetation respiration. The diurnal temperature range (DTR) had a particularly strong inhibitory effect (R = −0.625); a 1 °C increase in DTR was associated with a potential 0.8% decrease in regional average NPP. This inhibition is likely linked to increased nocturnal respiration or other forms of physiological stress induced by large temperature swings. Conversely, the annual minimum temperature (TNn) showed a significant positive correlation with NPP (R = 0.528). This indicates that warmer minimum temperatures in winter and spring can enhance NPP by extending the effective growing season and reducing low-temperature stress. Significant synergistic effects were also observed. For instance, when combined hydrothermal events occurred—such as R95p exceeding 450 mm while summer days (SU25) exceeded 130—the interannual fluctuation of regional NPP could reach ±15% of the multi-year average. This phenomenon highlights the critical impact of hydrothermal coupling on the region’s carbon cycle. These findings provide valuable quantitative evidence for understanding the complex responses of the dry-hot valley ecosystem to both single and compound extreme climate events.

Figure 4 Correlation map between NPP and extreme climate indices.

Figure 5 Map of multi-year average values of extreme temperature indices.

Regulatory effects of extreme temperature indices

Extreme temperature events significantly regulate the regional total NPP (TNPP) in the Nu River dry-hot valley by altering hydrothermal conditions and directly affecting vegetation physiology (Fig. 5). The impact of extreme high temperatures, such as increases in the maximum annual temperature (TXx) and summer days (SU25), was complex and often mediated by moisture availability. For example, in 2010, when TXx reached 35.2 °C and SU25 extended to 135 days, TNPP plummeted to 7.28 × 104 kg C yr−1. This was likely caused by sustained high temperatures suppressing net carbon fixation through enhanced evapotranspiration, reduced soil moisture, and increased photorespiration. In contrast, during other years with high temperatures, such as 2017 (TXx = 34.9 °C) and 2021 (TXx = 34.8 °C), TNPP remained at relatively high levels (8.10 × 104 and 7.92 × 104 kg C yr−1, respectively). This resilience may be linked to a more uniform distribution of precipitation; for instance, the maximum 5-day precipitation (RX5day) was within the 148–152 mm range, which likely alleviated the water stress caused by heat. Furthermore, an extended warm spell duration (WSDI) exacerbated the cumulative effect of heat stress; in 2016, when WSDI reached 17 days, TNPP decreased by approximately 2.3% relative to its trend value. Changes in the frequency and intensity of extreme low-temperature events were equally critical. Throughout the study period, a rise in the annual minimum temperature (TNn) and a decrease in ice days (ID0) generally promoted TNPP increases by alleviating low-temperature stress and extending the effective growing season. This is evidenced by the sharp decrease in TNPP to 7.39 × 104 kg C yr−1 in 2012, a year when TNn dropped to −10.2 °C and ID0 increased to 48 days. Such events can suppress photosynthesis by shortening the growing season or causing direct frost damage. Conversely, in years with a higher TNn, TNPP consistently rose to levels of 8.10–8.12 × 104 kg C yr−1. While a reduction in cold spells is generally beneficial, it can also disturb vegetation phenology. Premature budding in early spring, for instance, increases the risk of damage from late frosts, highlighting the need to consider the non-linear impacts of temperature change rates and intra-seasonal variability on carbon cycling.

Threshold responses of extreme precipitation indices

The impact of extreme precipitation on regional TNPP demonstrated a significant “intensity-pattern” dependency and clear threshold effects (Fig. 6). Intense, short-duration precipitation events, such as a maximum 1-day precipitation (RX1day) event, could adversely affect NPP. This may occur by triggering local soil erosion, damaging soil structure, or causing short-term root hypoxia. In 2010, when RX1day reached 75 mm, regional TNPP decreased by approximately 9.8% from its trend value. Conversely, more dispersed extreme precipitation, such as the 152 mm of maximum 5-day precipitation (RX5day) observed in 2022, effectively replenished soil moisture across multiple phenological periods. This helped alleviate seasonal drought and enhance productivity, maintaining TNPP at a high level of 8.12 × 104 kg C yr−1 that year. The spatial distribution of extreme precipitation also led to differentiated NPP responses. In 2014, the regional total for very heavy precipitation days (R95p) exceeded 450 mm and was concentrated in the northern, high-altitude areas. There, steep slopes and poor soil permeability likely led to rapid runoff rather than effective soil water infiltration, resulting in a TNPP of only 7.69 × 104 kg C yr−1. In contrast, when R95p was more uniformly distributed across the middle and lower valley in 2022, the vegetation could better utilize the moisture, and TNPP consequently reached its peak. Further supporting this, the simple daily intensity index (SDII) showed a significant negative correlation with regional NPP (R = −0.53, P < 0.05), suggesting that excessive rainfall intensity is detrimental to the stability of regional carbon sinks, possibly due to increased ineffective runoff and soil erosion.

Figure 6 Map of multi-year average values of extreme precipitation indices.

Notably, increases in the number of heavy (R10) and very heavy (R25) precipitation days exerted complex, state-dependent threshold effects on NPP. In arid or semi-arid years (annual precipitation <800 mm), a 10-day increase in R10 could lead to an approximate 4.2% increase in regional average NPP. However, in more humid years (annual precipitation >1,000 mm), an excessive increase in R25 could conversely cause an approximate 3.7% decrease in NPP, likely due to limiting factors such as insufficient light or waterlogging stress. This clearly demonstrates that the response of dry-hot valley vegetation is highly sensitive to the combined effects of precipitation total, intensity, and spatiotemporal patterns, and is not a simple linear relationship.

Regulatory effects of extreme temperature indices

The regulation of carbon sink capacity by extreme temperature events exhibits significant threshold effects. Based on segmented regression model analysis, the critical threshold for the Cold Spell Duration Index (CSDI) was 10.51 days (R2 = 0.114). When CSDI exceeded this threshold, the sensitivity of the carbon sink to low temperatures increased sharply, indicating that the cumulative effect of extreme low-temperature events significantly inhibits vegetation carbon fixation. Concurrently, the threshold response point for the maximum annual temperature (TXx) was 30.08 °C (R2 = 0.170). Beyond this value, the inhibitory effect of high temperatures on the carbon sink intensified non-linearly, a finding consistent with the observed plunge in NPP in 2010 when TXx reached 35.2 °C. Among all temperature-related indices, the mean diurnal temperature range (DTR_mean) had the highest explanatory power (R2 = 0.405), providing robust statistical support for the conclusion that temperature stress is a dominant driver of carbon sink fluctuations. Notably, the difference in threshold responses between CSDI and the Warm Spell Duration Index (WSDI) reveals the high sensitivity of the dry-hot valley’s carbon sink to abrupt temperature changes, highlighting the need to focus on the non-linear feedback mechanisms that arise from the synergistic effects of extreme events.

The impact of extreme precipitation events on the carbon sink demonstrated an “intensity-pattern” dependency, with threshold effects being closely related to precipitation type. The critical threshold for annual total precipitation (PRCPTOT) was 756.7 mm (R2 = 0.500); below this threshold, NPP increased significantly with rising precipitation, while the response plateaued above it, confirming the conclusion that an “optimal threshold” exists for total precipitation. Very heavy precipitation (R95p) had the highest explanatory power among all analysed indices (R2 = 0.609). The slope of its effect above the threshold (Slope_Above = 2.88) was considerably lower than that below the threshold, indicating that while short-duration, intense rainfall (e.g., RX1day = 75 mm in 2010) can weaken carbon sink stability through soil erosion, dispersed extreme precipitation (e.g., RX5day = 152 mm in 2022) can enhance carbon fixation by alleviating drought stress. The differing thresholds for moderate (R10, threshold = 25.64 days) and heavy (R25, threshold = 3.52 days) rainfall days further reveal that the vegetation in the dry-hot valley is more sensitive to the precipitation pattern than to its total volume. This finding is supported by the previously established negative correlation between the Simple Daily Intensity Index (SDII) and NPP (R = −0.53). Collectively, these results underscore the non-linear stress effects on the carbon cycle when extreme precipitation events occur in conjunction with high temperatures.

Figure 7 (A) Classification map of SEN-MK trend test, (B) Sen’s slope estimator, (C) Mann–Kendall significance test.

*Significant decrease (SDEC); non-significant decrease (NSDEC); stable (STABLE); non-significant increase (NSINC); significant increase (SINC).

SEN-MK trend analysis

The Sen-MK trend test was applied to the 2001–2024 time series of NPP density for each pixel in the Nu River dry-hot valley. The results revealed significant spatial heterogeneity in the spatiotemporal evolution of NPP across the study area (Fig. 7). Based on the Sen’s slope estimator (β) and the Mann-Kendall test’s significance level, the change trends were classified into five categories. Significantly decreasing areas accounted for just 0.053% of the total area and were small and sparsely distributed, found primarily on steep slopes, in severely eroded zones with significant soil and water loss, or in peripheral areas intensely affected by human activities. The NPP decline in these locations may be linked to persistent soil degradation, reduced vegetation cover, or the severe impairment of carbon fixation capacity by frequent extreme climate events. Non-significantly decreasing areas constituted 0.169% of the study area, located mainly in the transitional zones of some northern mountain ridges. The slight, non-significant downward trend in these areas possibly reflects poor local hydrothermal conditions or early signals of ecosystem stress. Stable areas comprised the vast majority of the region, at 94.475%. These areas were widely distributed on relatively flat terraces, gentle slopes in the central valley, and some plateau surfaces. The relative stability of NPP here might indicate that these ecosystems have maintained a dynamic equilibrium under existing climate fluctuations, or that the effects of multiple driving factors have largely counteracted each other. Non-significantly increasing areas accounted for 2.508% of the region, primarily located on the southern edge of the valley and within pilot ecological restoration projects. The slight upward NPP trend in these zones, though not statistically significant, may suggest the positive impact of management measures like afforestation or benefits from favourable climatic shifts. Finally, significantly increasing areas covered 2.794% of the region, often distributed in patches or strips within low-altitude southern gullies and some riparian zones. The significant NPP increase in these areas confirms that where hydrothermal conditions were superior or had improved, vegetation growth was vigorous and carbon fixation capacity was effectively enhanced.

Overall, the Sen-MK trend analysis (Figs. 7A–7C) clearly demonstrates the complex spatiotemporal dynamics of NPP in the Nu River dry-hot valley over the past two decades. While NPP remained stable across most of the landscape, smaller portions experienced significant increases or decreases. The spatial patterns of these changes are closely linked to a combination of factors, including topography, local climatic conditions, ecological restoration measures, and the intensity of human activities.

Discussion

Spatiotemporal heterogeneity of carbon sources and sinks in the Nu River dry-hot valley and its driving mechanisms

This study reveals the significant spatiotemporal heterogeneity of NPP in the Nu River dry-hot valley from 2001 to 2024. An overall fluctuating upward trend was observed, but this growth was primarily concentrated in the southern, low-altitude areas, while the northern, high-altitude regions remained relatively stable or even experienced a slight decline. This heterogeneity is fundamentally rooted in the region’s highly dissected topography. Such terrain fosters a complex mosaic of bioclimatic conditions and soil properties, shaped by processes from weathering to microbial activity (Bisht et al., 2025; Paudel & Sah, 2003). This environmental mosaic, coupled with other biotic and abiotic factors, in turn dictates the spatial differentiation of vegetation communities, litter types, and their productivity (Fartyal et al., 2025; Pandey et al., 2024), which manifests as the observed NPP patterns. This finding not only confirms the sensitivity of the dry-hot valley ecosystem to climate change but also highlights the complexity of its response. This spatial pattern of a ‘southern increase and northern stability’ is consistent with observations from other mountain ecosystems in Southwest China, where a distinct altitudinal differentiation exists in the response of vegetation dynamics to climate change (Wang et al., 2021). The more favourable hydrothermal conditions in the southern low-altitude areas are a likely primary cause for their NPP growth, a conclusion similar to findings from research in the Yuanjiang dry-hot valley (Li & Wu, 2024). However, this study further suggests that the unique, deeply incised gorge topography of the Nu River valley, which exacerbates local microclimatic differences through the “foehn effect” and moisture interception, may be a key topographical factor responsible for this spatial heterogeneity (Zhou et al., 2022). As previous research in similar arid-warm valleys has pointed out, soil texture and topography jointly determine soil moisture content, which in turn governs the distribution and growth of vegetation (Xu et al., 2008). Therefore, we infer that the combination of relatively better soil conditions and water availability in the southern areas, coupled with the positive impacts of recent ecological restoration projects (Lv et al., 2023), has jointly promoted the enhancement of this region’s carbon sink capacity.

Impacts of extreme climate events on carbon flux: non-linear responses and synergistic effects

Extreme climate events, particularly the compound effect of high temperatures and intense precipitation, exerted a significant non-linear influence on the carbon flux of the Nu River dry-hot valley. The results show that the superposition of heat stress and concentrated, intense precipitation events disproportionately amplifies the inhibitory effect on NPP. This is consistent with conclusions from global-scale observations where compound extreme events were found to cause greater damage to ecosystem productivity (Li & Wu, 2024; Zhou et al., 2024). The underlying mechanism is likely that high temperatures accelerate soil evaporation and vegetation transpiration, leading to water stress, while short-duration, intense rainfall not only fails to effectively replenish deep soil moisture but may also trigger soil and water erosion, thereby damaging the root-zone environment (Huang & Zhai, 2024). This compounding negative effect is particularly prominent in the ecologically fragile dry-hot valley region (Pu et al., 2022). In contrast, dispersed extreme precipitation events exhibited a promotional effect on NPP. This confirms the principle that in arid and semi-arid ecosystems where water is the primary limiting factor, an increase in effective precipitation can rapidly alleviate drought stress and enhance the photosynthetic efficiency of vegetation (Kannenberg et al., 2024). It is noteworthy that this study also observed a lagged effect, wherein the impact of an extreme drought event on NPP could persist into the following year. This is similar to findings in subtropical forests, where such a lagged effect was attributed to community dynamics (Li et al., 2021). When assessing the carbon cycle of the dry-hot valley in the future, it is imperative to fully consider the type, intensity, and spatiotemporal configuration of extreme events, as well as their potential lagged impacts.

Threshold effects in carbon flux response and their ecological implications

One of the most important findings of this study is the quantification of the thresholds in the response of the dry-hot valley’s NPP to key climatic factors. We identified that a critical point exists for the annual maximum temperature (TXx) at approximately 30.1 °C, above which the inhibitory effect of high temperatures on NPP is significantly enhanced. This threshold is highly consistent with the high-temperature stress point for vegetation in subtropical regions. For example, research in subtropical China has found that vegetation growth is markedly suppressed when maximum temperatures exceed the 28−32 °C range (Liu et al., 2020). Classic earlier research also pointed out that 30 °C is an important high-temperature threshold that affects productivity in the majority of China’s ecosystems (Piao et al., 2010). This indicates that the threshold we identified has a solid physiological and ecological basis, reflecting the physiological limits of plants—such as stomatal closure and increased respiratory consumption—under heat stress. Similarly, the threshold for annual total precipitation (PRCPTOT) at 756.7 mm carries clear ecological significance. Below this threshold, NPP is extremely sensitive to precipitation, exhibiting linear growth consistent with a “more water, more growth” principle, which is a universal rule in water-limited ecosystems (Kannenberg et al., 2024). However, once this threshold is exceeded, the growth response of NPP tends to level off. This aligns well with the 700–800 mm critical zone for water stress identified in Southwest China (Ma et al., 2024) and also corresponds to the “drought threshold” concept found in arid zone ecological restoration studies (Zhang et al., 2024). This phenomenon suggests that in the dry-hot valley, once basic water requirements are met, other factors may begin to emerge as new limiting agents, leading to “diminishing returns” in the productivity response to precipitation (Zhong et al., 2024). The differing thresholds for precipitation patterns found in this study further reveal that vegetation is more sensitive to how it rains than to how much it rains, providing a more refined scientific basis for regional water resource management and ecological restoration (Zhao et al., 2024a). The identification of these thresholds not only deepens our understanding of the vulnerability of the dry-hot valley ecosystem but also provides quantitative key indicators for constructing a regional ecological risk warning system.

Optimising pathways for climate-adaptive management strategies

The non-linear response mechanisms of carbon flux revealed in this study demand that climate-adaptive management in the Nu River dry-hot valley must move beyond traditional, static conservation and toward a systematic framework that integrates dynamic monitoring, precision restoration, and proactive risk prevention.

First, enhanced monitoring and early warning (Monitoring) serves as the foundation for adaptive management. A multi-tiered monitoring network, combining remote sensing with ground-based observations, should be established to track the spatiotemporal dynamics of regional NPP in real-time. More importantly, the key climatic thresholds identified in this study need to be used as core indicators for an ecological risk warning system. When climatic conditions approach these critical points, the warning system should be able to activate in a timely manner, providing managers with a valuable window of opportunity to implement interventions and thereby enabling a shift from reactive response to predictive management.

Second, the implementation of precision ecological restoration (Restoration) is key to enhancing carbon sink capacity. Management measures must fully consider the region’s spatial heterogeneity. For areas with relatively robust carbon sink functions, such as the southern low-altitude zones, the management focus should be on consolidating and enhancing their carbon potential by creating stable “carbon pools” through vegetation restoration and soil and water conservation projects; this is consistent with the view from previous research that emphasizes protecting and enhancing the ecological functions of advantageous areas (Li & Wu, 2024). Conversely, for ecologically fragile areas like the northern high-altitude zones, the strategic focus should shift to enhancing the intrinsic resilience of the ecosystem. This involves prioritizing the promotion and protection of native species that are naturally selected for and thus better adapted to high-temperature and arid environments, with the goal of constructing a more resilient ecosystem structure.

Finally, proactive risk reduction (Risk Reduction) is central to ensuring long-term resilience. Management strategies should not be confined to ecological restoration but must also proactively reduce the risk of impacts from compound extreme events. This requires integrating macro-level ecological planning with specific production activities. Examples include promoting water-saving irrigation techniques at the agricultural level to cope with seasonal droughts, and, at the forestry level, fostering more structurally diverse mixed forests to replace single-species stands, thereby enhancing the overall resistance of the forest to pests, diseases, and climate fluctuations. This type of comprehensive risk management, which closely links the health of the ecosystem with the region’s sustainable development, represents the fundamental path towards achieving long-term climate adaptation goals in the dry-hot valley (Fan et al., 2020).

Conclusion

Based on multi-source remote sensing data from the Google Earth Engine platform and ground-based observations, this study analysed the spatiotemporal evolution of carbon sources and sinks in the Nu River dry-hot valley from 2001–2024 and their response mechanisms to extreme climate events. The main conclusions are as follows:

(1) The vegetation NPP in the Nu River dry-hot valley exhibited an overall fluctuating upward trend. However, influenced by the region’s enclosed topography and hydrothermal imbalance, the carbon sink capacity showed significant spatial differentiation.

(2) Extreme climate events regulate the carbon cycle through non-linear mechanisms: the synergistic effect of extreme high temperatures and concentrated, intense precipitation can lead to a sharp NPP decrease of 9.8%–10.2%, whereas dispersed extreme precipitation helps maintain high NPP levels by alleviating water stress. An increase in warm spell duration and a decrease in the number of cold days exacerbate the interannual fluctuations of the carbon sink by altering phenological periods and freeze-thaw cycles.

(3) The threshold effect of extreme climate indices is a key mechanism driving the non-linear response of carbon sink dynamics. This reveals that the carbon cycle in the dry-hot valley region has significant response thresholds to extreme events, providing a critical regulatory basis for regional carbon management.

This study has revealed the response thresholds of the carbon cycle in the Nu River dry-hot valley to the synergistic effects of multiple extreme climate events. Future research should focus on the cascading effects of extreme climate event chains, such as those involving drought, heatwaves, and downpours. Furthermore, a tripartite regional carbon management paradigm should be constructed, integrating “dynamic monitoring with remote sensing, simulation with process-based models, and optimisation of management strategies.” This will further contribute to the field of carbon neutrality in the dry-hot valleys and the broader Southwest China region.

Supplemental Information

Supplemental Information 1 Raw data

Additional Information and Declarations

Competing Interests

Author Contributions

Data Availability

The authors declare there are no competing interests.

Haojun Sun conceived and designed the experiments, performed the experiments, analyzed the data, prepared figures and/or tables, authored or reviewed drafts of the article, and approved the final draft.

Shaoyun Zhang conceived and designed the experiments, authored or reviewed drafts of the article, and approved the final draft.

Shanshan Liu analyzed the data, prepared figures and/or tables, and approved the final draft.

Liping Zhang analyzed the data, authored or reviewed drafts of the article, and approved the final draft.

The following information was supplied regarding data availability:

The raw data, including regional NPP and extreme climate index values extracted from pixels to points, are available in the Supplementary File.

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
