# Peer review of "Investigation of spatiotemporal evolution and driving mechanisms of carbon sources and sinks in dry-hot valleys under extreme climate events: a case study of the Nu River dry-hot valley"

_PeerJ, doi:10.7717/peerj.20268_

## Round 0.1 · original submission · Major Revisions

· Academic Editor

Major Revisions

The last section regarding managerial implications need improved organization. At present, it resembles a compilation of suggestions. Organizing actions into overarching thematic categories, such as “Monitoring,” “Restoration,” and “Risk Reduction,” would improve clarity and effectiveness.

**Language Note:** The review process has identified that the English language must be improved. PeerJ can provide language editing services - please contact us at [email protected] for pricing (be sure to provide your manuscript number and title). Alternatively, you should make your own arrangements to improve the language quality and provide details in your response letter. – PeerJ Staff

Reviewer 1 ·

Basic reporting

• A professional language edit is strongly recommended, as the manuscript suffers from numerous language and grammatical issues. In particular, long and convoluted sentences, especially in the abstract and introduction, impair readability and should be revised for clarity.

Experimental design

• The threshold regression approach is central to the paper’s contribution, yet the methodology is poorly explained. For example, the criteria for selecting specific thresholds (e.g., 30.1 °C for TXx, 862.8 mm for PRCPTOT) are not described in sufficient detail. Additionally, it is unclear whether the threshold models were fit using piecewise linear regression, segmented regression, or quantile regression.

• The use of inverse distance weighting (IDW) for spatial interpolation of climate indices is debatable. Geostatistical methods such as kriging may be more appropriate. Therefore, the authors should justify the choice of IDW and discuss any potential spatial biases introduced by this method.

• The study employs pixel-wise Pearson correlation to examine the relationship between NPP and climate indices. However, this approach assumes linearity, normality, and homoscedasticity, assumptions that are rarely met in ecological or remote sensing data. A major concern is that the analysis does not account for temporal autocorrelation or spatial dependency among neighbouring pixels. Furthermore, no correction for multiple comparisons is applied, increasing the risk of false positives when analysing numerous pixels and variables.

Validity of the findings

No cross-validation or independent dataset appears to have been used to validate either the NPP trends or the threshold models, which weakens the robustness of the findings.

The claim that the study “validates the sensitivity of the regional carbon cycle to critical transitions” appears overstated. No dynamic modelling or feedback mechanisms are explicitly simulated to support this assertion.

Additional comments

• The final section on management implications should be better organised. As it currently stands, it reads more like a list of recommendations. Grouping actions under broader thematic categories, such as “Monitoring,” “Restoration,” and “Risk Reduction”, would enhance clarity and impact.

Reviewer 2 ·

Basic reporting

-

Experimental design

-

Validity of the findings

-

Additional comments

The topic is relevant and well-aligned with the scope of PeerJ. The research is methodologically sound and addresses important cause-effect relationships between environment, biodiversity, and society. The title accurately reflects the content, and the findings are of interest.
However, the manuscript requires revisions to improve clarity and scientific rigor:
- Revise the introduction to better establish the current state of knowledge.
- Clearly state the research hypotheses in the final paragraph of the introduction.
- Substantially revise the Results and Discussion section; avoid repeating results in the discussion.
- Update and expand the citations to include recent literature.
- Improve the overall quality of English for clarity and readability.
Please revise the manuscript accordingly.

---

## Round 0.2 · Minor Revisions

· Academic Editor

Minor Revisions

Needs revision as suggested in the reviewed manuscript.

Reviewer 2 ·

Basic reporting

The topic of the paper entitled “Investigation of spatiotemporal evolution and driving mechanisms of carbon sources and sinks in dry-hot valleys under extreme climate events: a case study of the Nu River dry-hot valley” is noteworthy and falls within the scope of PeerJ. In this manuscript, the authors have evaluated the variations in carbon flux along the communities using the scientific/technical methods. The research topic and data collected are intrinsically interesting because such types of the studies impact the cause-effect relationships between environment, biodiversity, ecosystems as well as society. The research topic and findings are inspiring. Overall, the title clearly reflects the contents. The following suggestions may help to improve the quality of the manuscript further.
The introduction does establish the existing state of knowledge but needs minor revision as suggested in the reviewed manuscript.
Authors should add the Hypotheses in the last para of Introduction.
Methodology needs revision as suggested
Results and Discussion part needs revision as suggested in the text.
Please avoid repeating the results in Discussion.
The citations need up-dated.
English needs minor improvement.
I have made more corrections and suggestions directly in the manuscript.
Please revise the paper accordingly.

Experimental design

Appropriate

Validity of the findings

Appropriate

Additional comments

I have made more corrections and suggestions directly in the manuscript.
Please revise the paper accordingly.

Annotated reviews are not available for download in order to protect the identity of reviewers who chose to remain anonymous.

---

## Round 0.3 · accepted · Accept

· Academic Editor

Accept

This revised version is suitable for publication in PeerJ.

Reviewer 2 ·

Basic reporting

The paper has been revised as per suggestions made by reviewers,
Authors- Please check the reference section as some of the references are incomplete (Volume/page No. are missing) e.g.,
Bisht V, Sharma S, Bargali SS, and Fartyal A. 2025. Topographic and Edaphic Factors Shaping Floral Diversity Patterns and Vegetation Structure of Treeline Ecotones in Kumaun Himalaya. Land Degradation & Development.36(12):4260-4280
Please check it very thoroughly.
I am satisfied with the revised version.
May be accepted for publication.

Experimental design

Appropriate

Validity of the findings

Appropriate

Additional comments

Revised version May be accepted.